# Forecast of myocardial infarction incidence, events and prevalence in England to 2035 using a microsimulation model with endogenous disease outcomes

**Peter Scarborough**[1,2]*, **Asha Kaur**[1], **Linda J. Cobiac**[1,3]

1 Nuffield Department of Population Health, University of Oxford, Oxford, United Kingdom, 2 National Institute of Health and Care Research Biomedical Research Centre at Oxford, Oxford, United Kingdom, 3 Griffith University, Queensland, Australia

* peter.scarborough@ndph.ox.ac.uk

## Abstract

### Background

Models that forecast non-communicable disease rates are poorly designed to predict future changes in trend because they are based on exogenous measures of disease rates. We introduce microPRIME, which forecasts myocardial infarction (MI) incidence, events and prevalence in England to 2035. microPRIME can forecast changes in trend as all MI rates emerge from competing trends in risk factors and treatment.

### Materials and methods

microPRIME is a microsimulation of MI events within a sample of 114,000 agents representative of England. We simulate 37 annual time points from 1998 to 2035, where agents can have an MI event, die from an MI, or die from an unrelated cause. The probability of each event is a function of age, sex, BMI, blood pressure, cholesterol, smoking, diabetes and previous MI. This function does not change over time. Instead population-level changes in MI rates are due to competing trends in risk factors and treatment. Uncertainty estimates are based on 450 model runs that use parameters calibrated against external measures of MI rates between 1999 and 2011.

### Findings

Forecasted MI incidence rates fall for men and women of different age groups before plateauing in the mid 2020s. Age-standardised event rates show a similar pattern, with a non-significant upturn by 2035, larger for men than women. Prevalence in men decreases for the oldest age groups, with peaks of prevalence rates in 2019 for 85 and older at 25.8% (23.3–28.3). For women, prevalence rates are more stable. Prevalence in over 85s is estimated as 14.5% (12.6–16.5) in 2019, and then plateaus thereafter.

**Data Availability Statement:** All data used for this paper were requested from public repositories,

sometimes with conditions attached to access. These repositories include: The UK Data Archive, which stores data from the Health Survey for England (HSE) series (https://www.data-archive.ac.uk/). Specifically, we use data from the following surveys (DOI for UK Data Archive storage location provided in brackets): • HSE1993 (DOI: 10.5255/UKDA-SN-3316-1) • HSE1994 (DOI: 10.5255/UKDA-SN-3640-2) • HSE1995 (DOI: 10.5255/UKDA-SN-3796-2) • HSE1996 (DOI: 10.5255/UKDA-SN-3886-2) • HSE1997 (DOI: 10.5255/UKDA-SN-3979-2) • HSE1998 (DOI: 10.5255/UKDA-SN-4150-1) • HSE1999 (DOI: 10.5255/UKDA-SN-4365-1) • HSE2000 (DOI: 10.5255/UKDA-SN-4487-1) • HSE2001 (DOI: 10.5255/UKDA-SN-4628-1) • HSE2002 (DOI: 10.5255/UKDA-SN-4912-1) • HSE2003 (DOI: 10.5255/UKDA-SN-5098-1) • HSE2004 (DOI: 10.5255/UKDA-SN-5439-1) • HSE2005 (DOI: 10.5255/UKDA-SN-5675-1) • HSE2006 (DOI: 10.5255/UKDA-SN-5809-1) • HSE2007 (DOI: 10.5255/UKDA-SN-6112-1) • HSE2008 (DOI: 10.5255/UKDA-SN-6397-2) • HSE2009 (DOI: 10.5255/UKDA-SN-6732-2) • HSE2010 (DOI: 10.5255/UKDA-SN-6986-3) • HSE2011 (DOI: 10.5255/UKDA-SN-7260-1) • HSE2012 (DOI: 10.5255/UKDA-SN-7480-1) • HSE2013 (DOI: 10.5255/UKDA-SN-7649-1) • HSE2014 (DOI: 10.5255/UKDA-SN-7919-3) • HSE2017 (DOI: 10.5255/UKDA-SN-8488-2) The Office for National Statistics mortality division, which provided data on deaths from myocardial infarction by age, sex and year (https://www.ons.gov.uk/peoplepopulationandcommunity/birthsdeathsandmarriages/deaths). This dataset is available from the University of Oxford Research Archive (DOI: 10.5287/bodleian:9eQ09708Z). All other data used in construction of the model were taken from published results from articles cited in the supporting information. The datasets compiling these results are available from the University of Oxford Research Archive (DOI: 10.5287/bodleian:9eQ09708Z). Full access to the R scripts that run the microPRIME model are available on GitHub (https://github.com/PeteScarbs/microPRIME/) and from the University of Oxford Research Archive (DOI: 10.5287/bodleian:9eQ09708Z).

**Funding:** PS was supported by a British Heart Foundation (www.bhf.org.uk) Intermediate Basic Science Research fellowship (FS/15/34/31656). The funders had no role in study design, data collection and analysis, decision to publish, or preparation of the manuscript.

**Competing interests:** The authors have declared that no competing interests exist.

## Conclusion

We may see an increase in event rates from MI in England for men before 2035 but increases for women are unlikely. Prevalence rates may fall in older men, and are likely to remain stable in women over the next decade and a half.

## Introduction

The burden of cardiovascular disease (CVD) within a population is a function of many underlying factors, including the demographics within the population (e.g. age distribution, ethnicity), the prevalence of behavioural risk factors (e.g. diet, physical activity, smoking) and related medical conditions (e.g. obesity, hypertension, diabetes), and the level of treatments available. In the UK, both incidence and death rates for CVD are falling [1], and have been since the 1960s [2], which must be due to some combination of changes in the underlying factors described above. The contributions of different factors to these trends have been studied both empirically and through modelling studies. A comprehensive analysis of incidence, case fatality and death rates from myocardial infarction (MI) using linked hospital episodes and death certificates in England [3] estimated that 57% and 52% of the reduction in death rates from men and women, respectively, between 2002 and 2010 were due to falls in event rates, and hence some combination of changes in behavioural risk factors. Unal et al. [4] used the IMPACT model to estimate that 58% of the modelled decline in coronary heart disease mortality in the UK between 1981 and 2000 was due to changes in population risk factors, with the rest due to improvements in treatment. The IMPACT model sums predicted reductions in mortality from calculations of population attributable risks for risk factors and treatments, and this method accounted for 89% of the reduction in CHD deaths over the time period studied.

At the end of the twentieth century, there were predictions of an 'obesity timebomb' [5], with adverse trends in both obesity and diabetes predicted to result in increases in CVD rates at some point in the future, and potentially a decline in life expectancy [6]. More recently, Pandya et al. [7] forecast that "continued improvements in cardiovascular disease treatment and declining smoking rates will not outweigh the influence of increasing population age and obesity on cardiovascular disease risk". To date, increases in CVD rates in the UK have not been observed, but it is still unclear whether such increases are around the corner. This is because CVD forecasting models include a 'baseline' future CVD trend that is built around data on previous CVD rates. Most often, this baseline is an extrapolation of previous trends (e.g. [8–13]). Sometimes, the baseline assumes constant CVD rates (e.g. [14, 15]). Models that include projections of CVD rates as exogenous inputs (either extrapolation of trends or assumption of constant future rates) cannot predict a future change in trend in CVD rates as a result of the balance of conflicting trends in risk factors and treatments. To do so, future CVD rates must be entirely endogenous, and CVD rates should emerge from the model based entirely on inputs about population demographics, risk factors and treatment.

Recent developments in infectious disease modelling have provided tools for coping with the combined uncertainty of multiple factors over time. For example, Bayesian history matching approaches have been used to forecast the prevalence of HIV rates [16]. This method uses techniques developed in the fields of geology and astrophysics to develop models where trends in the outcome of interest (in this case, disease rates) are endogenous to the model. Models using this approach are based on a theory of disease development which relies on parameters that are not known with precision, and the uncertainty around these parameters creates large

uncertainty around the modelled disease rates. This uncertainty can be narrowed by calibrating the parameter space to those areas which produce modelled disease rates that are similar to measured estimates from an external dataset. If a model has a large number of uncertain parameters and a long run time, this process is usually assisted by model emulation (i.e. using training data to develop equation-based approximations of the full model). In this paper, we use this method to forecast myocardial infarction (MI) incidence, event and prevalence rates in England to 2035. As the disease rates produced by our model (microPRIME) are endogenous and based on trends in demographics, risk factors and treatments, the results demonstrate whether current adverse trends in obesity and diabetes can outweigh positive trends in blood pressure, cholesterol, smoking and treatment and produce an increase in disease rates in the next decade and a half.

## Materials and methods

A complete description of the microPRIME model is provided in the (S1 Text). In summary, the microPRIME model is a framework of interconnecting modules developed in the R coding language, including a microsimulation of MI events within a sample of 114,000 agents representative of the age and sex structure of England. For each agent in turn, microPRIME simulates a history over 37 annual time points, from 1998 to 2035. At each annual time point, the agent can have an MI event, die from an MI, or die from an unrelated cause based on a stochastic process with probability estimated as a function of the agent's age, sex and risk factor status. This risk factor status consists of three continuous variables—BMI, systolic blood pressure (SBP) and total cholesterol–and three binary variables–smoking, diabetes and previous MI. Aggregated outputs are used to estimate annual prevalence rates (proportion of agents alive who have had a previous MI), incidence rates (the number of first MI events per 100,000), and event rates (the number of any MI events per 100,000).

The model is calibrated to external estimates of incidence, prevalence and event rates between 1999 and 2011. Because of the long run time of the microsimulation model (3 days for 114,000 agents run over 18 core processors), this process is assisted by an emulator. To produce training data for the emulator, the microsimulation model runs 450 times, with each run using a different set of model parameters selected at random from their underlying distribution. The model parameters control the following elements of the microPRIME structure: relative risks linking risk factors and MI incidence; temporal trends in risk factors; and temporal trends in 30 day case fatality rates for MI (which is used as a proxy of treatment effects). The emulator uses the training data to estimate model outcomes over 2,000,000 vectors of parameters drawn at random from the total parameter space. Parameter vectors that produce model outcomes that are too far from externally measured estimates of incidence, prevalence and event rates are rejected. For incidence and event rates the external dataset is a linked dataset of hospital episodes and death certificates [3]. For prevalence, the external dataset is the Health Survey for England series (e.g. [17])–a representative sample of approximately 8,000 adults in England that has run annually since 1992 and has focussed on risk factors for and prevalence of cardiovascular disease in 1998, 2003, 2005, 2006, 2011 and 2017. A sample of 450 of the remaining sets of model parameters is then used in the microsimulation to produce more training data for a more refined set of emulators. This process is continued until pre-defined stopping criteria are achieved and a final set of 450 parameter vectors are obtained. These are used to run the microsimulation to 2035 to produce forecasts of MI incidence, prevalence and event rates. The 450 iterations used in our modelling was greater than the minimum threshold recommended for fitting emulators [16] (for further information about selecting the number of agents and iterations, see the 'emulator' section of the S1 Text).

We present forecasts of MI incidence and prevalence rates by sex for four age groups: 55–64, 65–74, 75–84 and 85+. We also present forecasts of event rates that have been age-standardised to the English over-55 age structure in 2018. Each set of estimates is accompanied by 95% uncertainty intervals, which include results between the 2.5th and 97.5th percentiles of 450 model runs. Trends in BMI, SBP, cholesterol, smoking, diabetes and treatment for MI continue in the forecast period (2017–2035) as observed in the period 1992–2016. Details about the BMI trends are published elsewhere [18]–we use future projections where mean BMI approaches an asymptote. Trends in the other risk factors and treatment variables are covered in the (S1 Text). The modelled trends in these risk factors are displayed in the S1 Text in two ways. S2 Fig in S1 Text shows the trends in each risk factor between 1998 and 2035 (mean for continuous variables, prevalence for binary variables). S2 Table in S1 Text compares modelled estimates for each risk factor for the year 2011 with estimates from the 2011 Health Survey for England, to demonstrate compatibility of the modelled trends with measured data at a single point in time.

We validate the model against age and sex-specific estimates of the prevalence of MI from the Health Survey for England 2017 [19]. These estimates were left out of the emulator's external dataset so as to be independent of the model building process. We considered the model to be validated if the 95% confidence intervals for the prevalence estimates overlapped with the 95% uncertainty intervals of the model after a Bonferroni adjustment for multiple comparisons [20].

## Results

Fig 1 shows the projected incidence rates for first MI by age group and sex. For all age and sex groups, incidence of MI fell up to the end of the calibration period (2011) and then continued to fall for a further ten years, before steadying from approximately the year 2020. For women in the age range 75–84, there was a small upturn in the incidence rate after 2020, increasing from 643 (429–876) new events per 100,000 to 666 (427–949) per 100,000 by the end of the forecast period. Similarly, there was a small upturn for men aged 55–64, increasing from 463 (325–640) per 100,000 in 2019 to 592 (395–813) per 100,000 in 2035, but both of these increases were well within the 95% uncertainty intervals. Direct comparisons between incidence rates for men and women are shown in the S1 Text.

Modelled age-standardised event rates for MI (first and subsequent) are shown in Fig 2. Similar to the modelled results for incidence the model produces falling event rates across the calibration period, which extend to 2020. For women, event rates then show a very modest increase from 464 (365–583) per 100,000 in 2019 to 482 (351–666) per 100,000 in 2035. For men, rates increase more sharply from 845 (655–1038) per 100,000 in 2019 to 1042 (761–1371) per 100,000 in 2035. As for forecast incidence rates, these increases did not exceed 95% uncertainty intervals.

Modelled prevalence rates are shown in Fig 3. For men, there are diverging patterns of prevalence of ever having had a MI for older and younger age groups during the calibration period. Rates decrease for the 55–64 and 65–74 year olds, but increase for the 75–84 and 85+ age groups. In the forecast period we see a clear change in trend for all of the age groups. There is a peak of prevalence rates in 2019 for 85 and older at 25.8% (23.3–28.3), but then falls in prevalence rates after this. For the younger age groups, the falling prevalence rates plateau and then begin to increase in the mid 2020s. For women, prevalence rates appear to be more stable over both the calibration and forecast period. For the two younger age groups, prevalence rates slowly declined over the calibration period and continue to decline after this. For the over 85s, prevalence rates climb from 11.0% (10.4–11.7) in 1999 to 14.5% (12.6–16.5) in 2019, and then

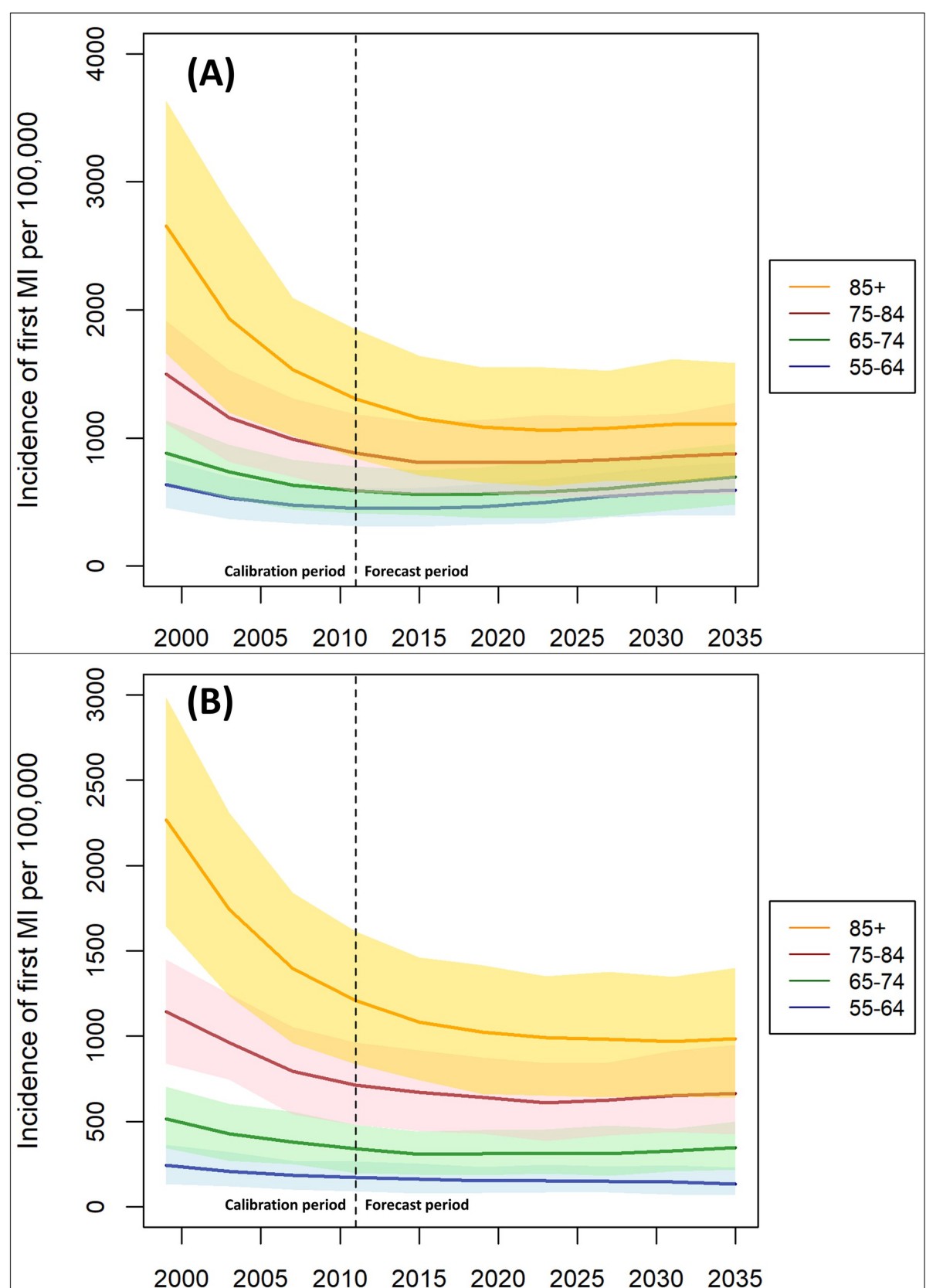

**Fig 1.** Modelled incidence rates of first MI by age group for men (A) and women (B), 1998–2035.

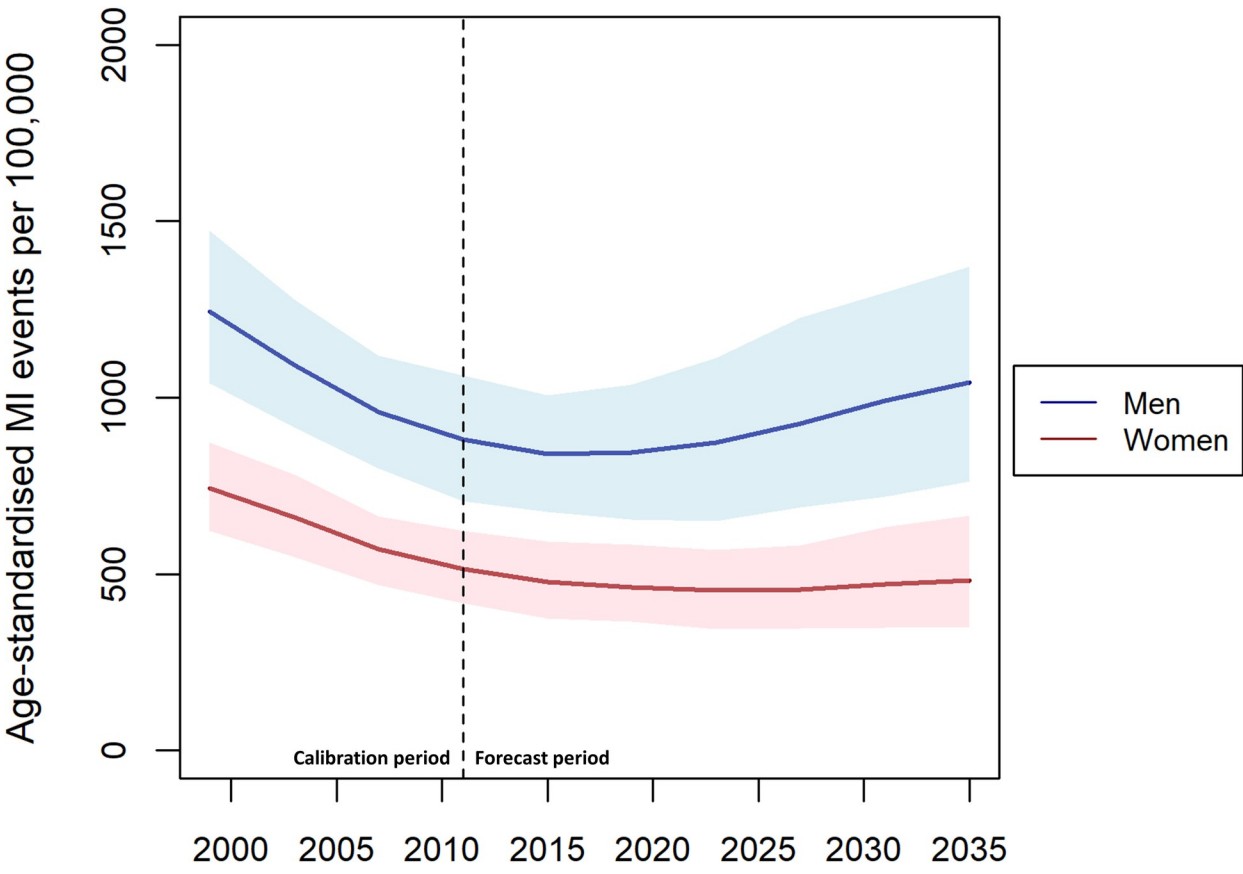

**Fig 2. Modelled age-standardised event rates for MI (first and subsequent) for men and women, over 55s only, 1998–2035.**

plateau. Prevalence rates are consistently higher for men than for women across all age groups (see S1 Text).

To validate the model we compared modelled estimates of the prevalence of ever having had MI in 2017 (six years after the calibration period) with external estimates from the Health Survey for England, and the results are shown in Fig 4. For each age-sex group, the 95% confidence intervals from the external dataset overlapped with the 95% uncertainty intervals from the modelled runs, with the exception of the 55–64 year old females. In this age-sex group, the modelled prevalence of MI in 2019 was 2.07% (1.77–2.38), compared to the external dataset estimate of 0.81% (0.01–1.73).

## Discussion

### Summary and implications

Using our unique model infrastructure, calibrated using methods that have been successfully applied to infectious disease modelling [16], we forecast that the adverse trends in obesity and diabetes that have been observed in the UK for the last thirty years may result in increases in MI incidence and events for men by 2035 but are unlikely to do so for women. The negative health consequences of these trends are balanced by improvements in smoking, blood pressure and cholesterol management, and reductions in 30-day case fatality linked to improved treatment. Due to improvements in survival from MI alongside reductions in incidence, we model increases in the prevalence of older men and women having had a previous MI until approximately 2020, and then steady levels for women and reductions in prevalence for men.

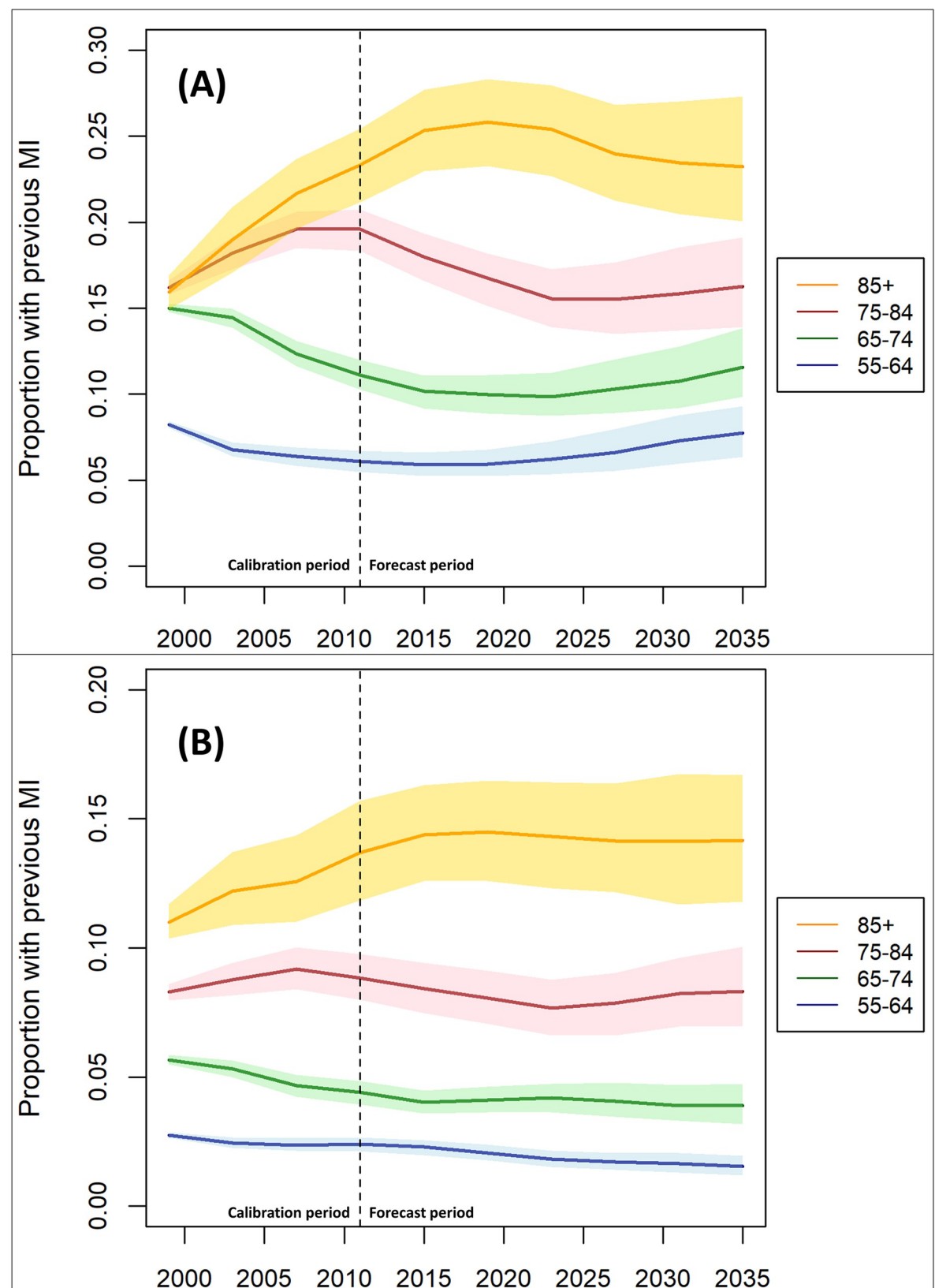

**Fig 3.** Modelled prevalence rates of first MI by age group for men (A) and women (B), 1998–2035.

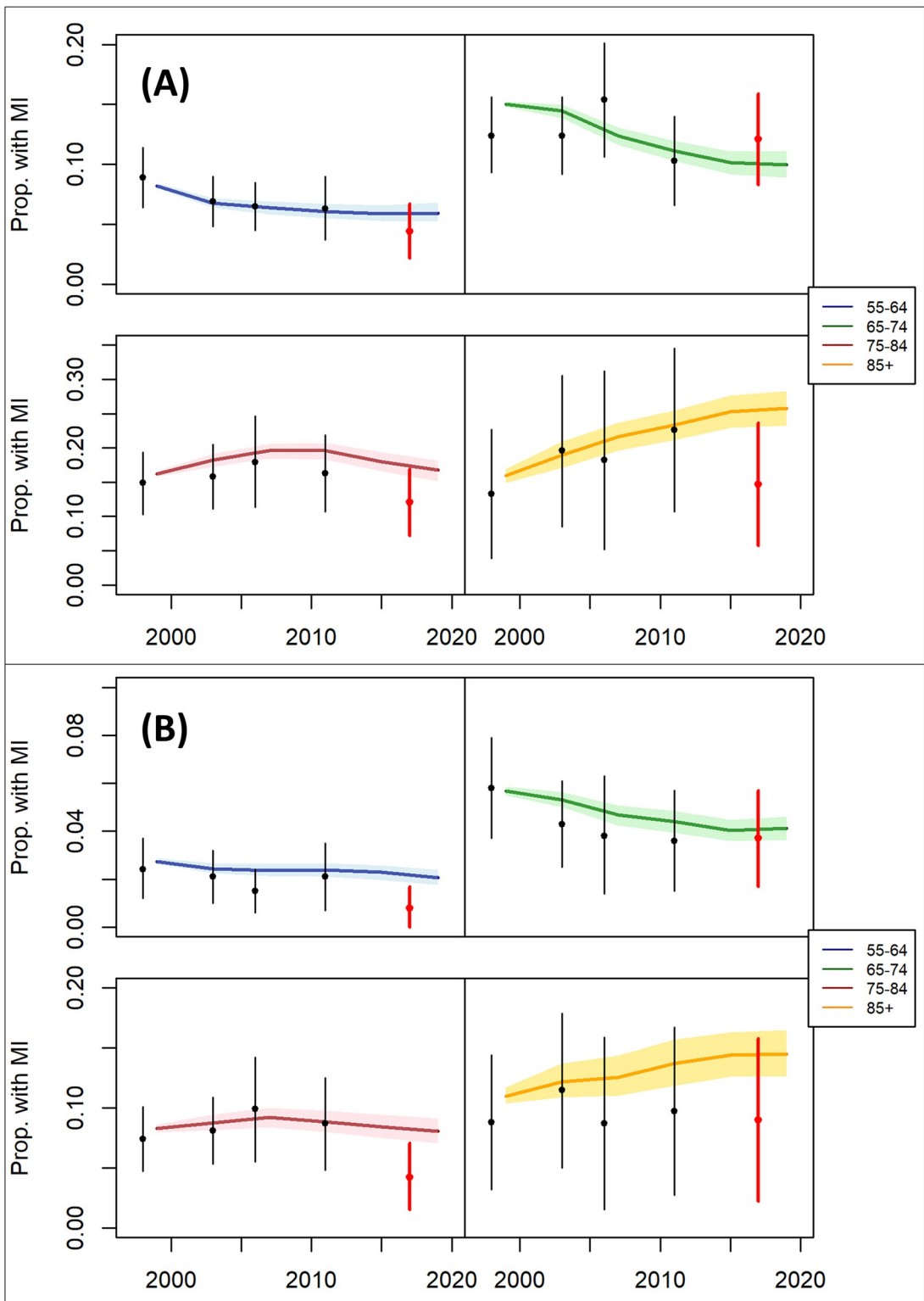

**Fig 4.** Validation of the model by comparison of modelled estimates of prevalence of ever having had MI with external estimates from the Health Survey for England 2017 for men (A) and women (B). Black vertical bars represent 95% confidence intervals for the prevalence of ever having had MI for 1998, 2003, 2006 and 2011, which were used in the model parameterisation process. Red vertical bars represent 95% confidence intervals for prevalence in 2017, which was not used in the model parameterisation process.

Although we only modelled MI rates, due to similar risk factors for other major cardiovascular diseases (angina, heart failure and stroke) our results suggest that large increases in cardiovascular disease incidence in England for women are not likely before 2035. They also suggest that increases in prevalence for both older men and women may have already occurred and will now be followed by declines in men and steady levels for women, which has implications for NHS resource planning.

We have demonstrated that forecast and scenario models for non-communicable diseases can be developed that do not rely on exogenous estimates of future disease rates. These methods could be applied to multiple diseases and risk factors to aide resource planning. By incorporating estimates of disease burden and economic costs, such models could be used to produce health economic estimates for evaluations of population-level health interventions and policies. Although modelling frameworks currently exist for that purpose [8, 21], the model structure introduced here builds on previous work by explicitly incorporating the effect of risk factors and treatment into observed and forecast trends in disease outcomes. There are two developments that would increase the usefulness of non-communicable disease forecast models, such as microPRIME. First, forecast models would benefit from multiple disease outcomes across different categories of non-communicable diseases. Being able to forecast across multiple diseases has obvious benefits for resource planning, but it could also improve the accuracy of model forecasts for any given disease. This is because non-communicable diseases share common risk factors, so trends in these risk factors will affect disease incidence across multiple diseases–without explicitly incorporating these multiple disease pathways in the model framework a model will not account for related trends in these competing risks. Second, public health policymakers would benefit from models that incorporate more behavioural risk factors for disease (e.g. poor diet, lack of physical activity, alcohol consumption). By including such risk factors (either through direct associations with disease outcomes, or indirectly via body weight, blood pressure etc.) forecast models would be able to run policy scenarios aimed at changing behaviour.

## Strengths and limitations

The microPRIME model is a unique contribution to forecast and scenario modelling for non-communicable diseases. It has been validated against external estimates of the prevalence of ever having had a MI. The underlying code for microPRIME is freely available from https://github.com/PeteScarbs/microPRIME/ and the Oxford Research Archive (DOI: 10.5287/bodleian:9eQ09708Z). Although microPRIME currently operates for only a single disease, because it has been designed as a microsimulation adding new risk factors and disease outcomes is straightforward and should have only a small impact on computing time [22].

Although all forecasts of MI rates in microPRIME are endogenous, it does have to rely on exogenous forecasts of trends in risk factors and treatment which are of course inherently uncertain. Our extrapolations of trends in overweight and obesity [18] have demonstrated how different models can have extremely similar fit to past trends and still diverge considerably in forecasts. This shows how even when purportedly following a data-driven approach, the choices of modellers can impact on future predictions. Ideally, such choices could be explored in sensitivity analyses, and the range of potential results can be incorporated in uncertainty ranges around forecasts. For this paper, the uncertainty ranges were generated from the 2.5th and 97.5th percentiles of model outcomes from 450 model runs, and will be an underestimate of the true uncertainty of future MI rates as each model run assumes that past trends in risk factors and treatments will continue over the forecasted period.

The emulator that was built to estimate model outcomes across a wide range of the available parameter space was tested by assessing the correlation between emulated model outcomes and observed model outcomes (from training data generated by microPRIME model runs). These diagnostic tests revealed that the emulator was more successful at reproducing modelled estimates of MI prevalence and event rates than for incidence rates, and therefore the calibration process (where parameter space that produces results that are not compatible with observed outcomes from external datasets is removed) was weighted more towards prevalence and event rate outcomes.

The agents in the microPRIME model could die from a cause unrelated to MI, with probability based upon death rates for all causes where MI was not mentioned at any point on the death certificate. These death rates were included as exogenous inputs in both the calibration and forecast ranges of the model. The non-MI death rates were assumed to be unrelated to the risk factors included in the modelling process (smoking, blood pressure, cholesterol, BMI and diabetes), which is an over-simplification since these risk factors are related to many health outcomes other than MI. This implies that agents that had a high risk profile (and therefore are more likely to have an MI) would have death rates that are lower than would be expected in real life, which would result in an over-estimation of the prevalence of MI.

## Comparison with existing literature

In this paper we forecast future CVD rates for a population based on competing trends in treatment and risk factor status. Pandya et al. [7] forecast that CVD prevalence rates in the US would increase up to 2030 due to increasing obesity outweighing declines in smoking and improvements in treatment. However, their forecasts are partially based on exogenous trends in CVD rates. This is also the case for other CVD forecast models [23–27] and estimates of future life expectancy [28] or health service use [29]. Some models of population CVD outcomes base their forecasts on individual-level disease prediction scores that have been calibrated using historical disease rates in the population of interest (e.g. [30–32]). These models are the closest in design to microPRIME, but they do not allow for independent uncertainties around the relationships between the disease outcomes and the individual risk factors included in the risk prediction scores. Therefore, it is assumed that these relationships are the same as were observed in the cohort used to develop the risk score, despite requirement to calibrate the modelled disease outcomes from the risk scores to levels observed in the population under investigation. In microPRIME, all risk factors are included independently, and their joint relationships with MI incidence are those that are selected in the calibration process. The two most similar modelling approaches of which we are aware are the IMPACT$_{NCD}$ microsimulation model [33], and the Future Elderly Model [34]. The IMPACT$_{NCD}$ model generates static coronary heart disease incidence rates by applying population attributable fractions for blood pressure, cholesterol, smoking, body weight, physical activity, fruit and vegetable consumption and area-level deprivation. For a baseline calibration year, this approach estimates the incident rate expected if all risk factors were at the level associated with lowest risk, and this static incident rate is then used throughout the forecast modelling period. Our approach builds on this by using the history matching process to ensure the static rates are consistent with multiple historical estimates of MI incidence, event and prevalence rates. The Future Elderly Model projects a microsimulation representing the US population of adults aged 50 and over to forecast mortality rates, disease burden and healthcare costs. Forecasted mortality rates are endogenous to the model, and based on trends in sociodemographic factors, smoking status and disease conditions. We use a similar approach but apply it to disease incidence rather than total mortality.

## Conclusion

Using a modelling approach unique to non-communicable disease scenario models we forecast that if current trends in risk factors and treatment continue, we are unlikely to see an increase in incidence and event rates from MI in England for women before 2035, but incidence and events may begin to rise for men. The prevalence of having had a MI is likely to remain stable for women over the next decade and half. For older men, prevalence rates are likely to initially fall before plateauing.

## Supporting information

**S1 Text. Detailed description of the microPRIME model.**
(DOCX)

## Author Contributions

**Conceptualization:** Peter Scarborough.

**Data curation:** Peter Scarborough, Asha Kaur.

**Formal analysis:** Peter Scarborough, Linda J. Cobiac.

**Funding acquisition:** Peter Scarborough.

**Investigation:** Peter Scarborough, Asha Kaur, Linda J. Cobiac.

**Methodology:** Peter Scarborough, Asha Kaur, Linda J. Cobiac.

**Project administration:** Peter Scarborough.

**Resources:** Peter Scarborough.

**Software:** Peter Scarborough, Linda J. Cobiac.

**Supervision:** Peter Scarborough.

**Validation:** Peter Scarborough, Linda J. Cobiac.

**Visualization:** Peter Scarborough, Asha Kaur.

**Writing – original draft:** Peter Scarborough.

**Writing – review & editing:** Peter Scarborough, Asha Kaur, Linda J. Cobiac.

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
