## [Decision Letter · Decision Letter 0]

28 Mar 2022

PONE-D-21-32355Forecast of myocardial infarction incidence, events and prevalence in England to 2035 using a microsimulation model with endogenous disease outcomesPLOS ONE

Dear Dr. Scarborough,

Thank you for submitting your manuscript to PLOS ONE. After careful consideration, we feel that it has merit but does not fully meet PLOS ONE’s publication criteria as it currently stands. Therefore, we invite you to submit a revised version of the manuscript that addresses the points raised during the review process.

We look forward to receiving your revised manuscript.

Kind regards,

Yiqiang Zhan

Academic Editor

PLOS ONE

“PS is supported by a British Heart Foundation Intermediate Basic Science fellowship (FS/15/34/31656).”

“PS was supported by a British Heart Foundation (www.bhf.org.uk) Intermediate Basic Science Research fellowship (FS/15/34/31656). The funders had no role in study design, data collection and analysis, decision to publish, or preparation of the manuscript.”

Additional Editor Comments:Codes of the analysis could be supplemented in a separate file.

Reviewers' comments:

Reviewer's Responses to Questions

**Comments to the Author**

1. Is the manuscript technically sound, and do the data support the conclusions?

Reviewer #1: Partly

Reviewer #2: Yes

2. Has the statistical analysis been performed appropriately and rigorously? 

Reviewer #1: Yes

Reviewer #2: Yes

3. Have the authors made all data underlying the findings in their manuscript fully available?

Reviewer #1: No

Reviewer #2: Yes

4. Is the manuscript presented in an intelligible fashion and written in standard English?

Reviewer #1: Yes

Reviewer #2: Yes

5. Review Comments to the Author

Reviewer #1: This paper describes a microsimulation model of myocardial infarction (MI) incidence that seeks to model this outcome fully endogenously. In other words, rather than relying on extrapolation of trends in MI, the model instead relies on inputs about population demographics, risk factors, and treatment. In spirit, this has a similar philosophical basis as the Future Elderly Model and its related microsimulations.

The model includes three continuous variables (BMI, systolic blood pressure, and total cholesterol) and three binary variables (smoking, diabetes, and previous MI). MI is then a function of these (plus demographic characteristics and the impact of technological advancement in treating MI), incorporating assumptions about the future trends of these predictor variables. Several sets of results are presented, including future incidence rates (by age group and gender), age-standardized MI events by gender, future prevalence of MI (by age group and gender), and external validation of future prevalence of MI compared to observed data at one point particular point in time (2017).

It is challenging to describe a complex microsimulation model clearly and I think the authors have done a good job here, with one important exception. The authors rely heavily on calibrating their model to historic data (1993-2014 for risk factors, I think). It would help build confidence in the model if the key endogenous variables used in the simulation were summarized at a particular point in time (perhaps 2011?). For example, a table conveying statistics on BMI, systolic blood pressure, total cholesterol, and prevalence of smoking and diabetes. The reader would then feel confident that the model was at a reasonable place in 2011.

Similarly, it would help to provide a bit more on the extrapolated/fitted future trajectories in BMI, blood pressure management, cholesterol management, smoking, diabetes, and treatment for MI. These are areas where reasonable people likely differ in their assessment of the future, so some more information about these future trends, such as figures in the appendix would help greatly.

Figure 4 would be strengthened by including historic prevalence of MI from the survey data. This, too, will build credibility with the reader. I note that the PLOS ONE paper on BMI modeling (Figure 1 in https://doi.org/10.1371/journal.pone.0252072) includes historic data on BMI in a similar exercise.

Reviewer #2: This is an important simulation study to forecast myocardial infarction incidence, events and prevalence in England to 2035. I found the article well written and the methodological approach original and interesting. I do have some minor comments

1) Page 5: The authors should explain the choice of 450 model runs which is an unusual choice, usually you expect to see model runs closer to 10,000. The authors need to answer the following questions: Why did they choose 450 runs and not some other number? What is the expected impact of choosing a higher number of runs? I'm assuming that they chose the biggest number that was feasible in terms of estimation times? If that is the case I would have liked to see tests where they had a bigger numbers (than 450) of model runs for a smaller subset of agents or less horizon time to test how sensitive the model estimates as model runs increase for a given number of agents/horizon time.

2) Figures: Say what the dashed line means i.e. calibration year in all figures rather than just in the main text

3) Did you explore differences by age and gender rather than only separately? At the very least it would be useful to add a figure that is similar to Figure 1 but shows the different age groups by gender. In that way we can compare 85+ women vs men or 75-84 old women vs men and so on.

4) What is the policy recommendation from your research? If you do not have specific policy implications to add then what do you suggest for future research in this area? How can we improve our modelling forecasts? What kind of data do we need? Can you add something brief to the discussion section of the paper?

6. PLOS authors have the option to publish the peer review history of their article (what does this mean?). If published, this will include your full peer review and any attached files.

Reviewer #1: No

---

## [Author Response · Author response to Decision Letter 0]

27 May 2022

For responses to the editors, please see the 'Response to Reviewers' document. Responses to the reviewers are also provided below.

Reviewer 1 comments

This paper describes a microsimulation model of myocardial infarction (MI) incidence that seeks to model this outcome fully endogenously. In other words, rather than relying on extrapolation of trends in MI, the model instead relies on inputs about population demographics, risk factors, and treatment. In spirit, this has a similar philosophical basis as the Future Elderly Model and its related microsimulations.

Response: We had overlooked the Future Elderly Model, so we thank the reviewer for pointing this out. We agree that the microPRIME model uses similar methods to forecast health outcomes as the Future Elderly Model and we have added the following text to the ‘Comparison with existing literature’ section of the discussion: “The Future Elderly Model projects a microsimulation representing the US population of adults aged 50 and over to forecast mortality rates, disease burden and healthcare costs. Forecasted mortality rates are endogenous to the model, and based on trends in sociodemographic factors, smoking status and disease conditions. We use a similar approach but apply it to disease incidence rather than total mortality.”

The model includes three continuous variables (BMI, systolic blood pressure, and total cholesterol) and three binary variables (smoking, diabetes, and previous MI). MI is then a function of these (plus demographic characteristics and the impact of technological advancement in treating MI), incorporating assumptions about the future trends of these predictor variables. Several sets of results are presented, including future incidence rates (by age group and gender), age-standardized MI events by gender, future prevalence of MI (by age group and gender), and external validation of future prevalence of MI compared to observed data at one point particular point in time (2017). It is challenging to describe a complex microsimulation model clearly and I think the authors have done a good job here, with one important exception. The authors rely heavily on calibrating their model to historic data (1993-2014 for risk factors, I think). It would help build confidence in the model if the key endogenous variables used in the simulation were summarized at a particular point in time (perhaps 2011?). For example, a table conveying statistics on BMI, systolic blood pressure, total cholesterol, and prevalence of smoking and diabetes. The reader would then feel confident that the model was at a reasonable place in 2011.

Response: Thank you for your comments – we are very pleased that you find the methods section and the supplementary material to be a good description of the process. We agree with the reviewer that it would build confidence to present a comparison between modelled and measured estimates of risk factors in the year 2011. We have run these analyses and present the results in a new table in the supplementary material (table S2). We have also added a sentence to the methods section of the main paper to draw attention to this new table. From the 450 iterations of the microPRIME model we extracted estimates from 2011 of the mean of BMI, systolic blood pressure and total cholesterol, and prevalence of smoking and diabetes. There were 5 risk factors, 4 age categories and 2 sexes, resulting in 40 risk factor-age-sex groups for comparison with estimates from the Health Survey for England. For 34 of these groups, the uncertainty intervals from the microPRIME model overlapped with the 95% confidence intervals from the Health Survey for England data.

Similarly, it would help to provide a bit more on the extrapolated/fitted future trajectories in BMI, blood pressure management, cholesterol management, smoking, diabetes, and treatment for MI. These are areas where reasonable people likely differ in their assessment of the future, so some more information about these future trends, such as figures in the appendix would help greatly.

Response: We agree that this would help readers to understand how the model operates. We have added a new figure to the supplementary material that shows the trends in each of the risk factors over the periods for which the regression models were built (1993 – 2014), and projections of these trends to 2035. To demonstrate differences by sex and age, for each risk factor we display trends for men and women aged 40 and 80.

Figure 4 would be strengthened by including historic prevalence of MI from the survey data. This, too, will build credibility with the reader. I note that the PLOS ONE paper on BMI modeling (Figure 1 in https://doi.org/10.1371/journal.pone.0252072) includes historic data on BMI in a similar exercise.

Response: We have updated figure 4 by adding historic data on the prevalence of MI from 1998, 2003, 2006 and 2011. This demonstrates how are modelled estimates track the survey data over time.

Reviewer 2 comments

This is an important simulation study to forecast myocardial infarction incidence, events and prevalence in England to 2035. I found the article well written and the methodological approach original and interesting. I do have some minor comments

Response: Thank you for your helpful comments and review of the paper.

1) Page 5: The authors should explain the choice of 450 model runs which is an unusual choice, usually you expect to see model runs closer to 10,000. The authors need to answer the following questions: Why did they choose 450 runs and not some other number? What is the expected impact of choosing a higher number of runs? I'm assuming that they chose the biggest number that was feasible in terms of estimation times? If that is the case I would have liked to see tests where they had a bigger numbers (than 450) of model runs for a smaller subset of agents or less horizon time to test how sensitive the model estimates as model runs increase for a given number of agents/horizon time.

Response: We agree that the number of model runs used in this process is smaller than you would usually see for a health model. That is partly because of the high computational demands of the microPRIME model, and partly because estimates of uncertainty in the microPRIME model results are generated differently than for most health models. For microPRIME, we use emulators to reduce the set of parameter space that produces implausible results by calibration against external datasets (the ‘history matching’ approach described in the manuscript). It is this history matching approach that determines the size of the uncertainty intervals, rather than the number of model runs. However, the number of model runs is important as it (in part) determines the accuracy of the emulators. We did run tests to determine the number of model runs for optimal results with available computing power, and we have added the following text to the supplementary material to describe this process:

“For each of the model outcomes that are being emulated, one iteration of the microPRIME model provides one data point used to fit the emulators. For our analyses we used 450 iterations, so each emulator was fit using 450 data points. As a rule of thumb, Andrianakis et al. (2015) suggest there should be at least 10 data points for every parameter that is allowed to vary in the model. For our analyses we have 38 varying parameters, (19 for men and 19 for women). Since separate emulators are built for men and women, this means that we have over 23 data points per varying parameter. There are two methods to improve the fit of the emulators. First, you can increase the number of iterations in the training data – this provides a greater number of data points to fit the emulators. Second, you can increase the number of agents in each iteration – this reduces stochastic variation in model outcomes, which in turn reduces the random noise in the emulator fitting process. Both of these methods increase the time taken for both the microsim and emulator modules to run, so selecting appropriate numbers of agents and iterations requires a balance. We tested the microPRIME model over a range of 50,000-114,000 agents and 380-900 iterations, to observe the computation time and the number of emulated outcomes for which correlation with modelled outcomes achieved a minimum threshold (r > 0.6). We found that increasing the number of agents had a bigger impact on emulator fit than increasing the number of iterations. Therefore, we prioritised a high number of agents and moderate number of iterations in our final model.”

We have also added a small amount of text to the main paper, as follows:

“We used 450 iterations in our modelling as this was greater than the threshold recommended for fitting emulators [16] (for further information about selecting the number of agents and iterations, see the ‘emulator’ section of the supplementary material).”

2) Figures: Say what the dashed line means i.e. calibration year in all figures rather than just in the main text

Response: This has been added to all of the figures.

3) Did you explore differences by age and gender rather than only separately? At the very least it would be useful to add a figure that is similar to Figure 1 but shows the different age groups by gender. In that way we can compare 85+ women vs men or 75-84 old women vs men and so on.

Response: We have added new figures that demonstrate differences by sex in the supplementary material. 

4) What is the policy recommendation from your research? If you do not have specific policy implications to add then what do you suggest for future research in this area? How can we improve our modelling forecasts? What kind of data do we need? Can you add something brief to the discussion section of the paper?

Response: The primary aim of this paper is to introduce a model that uses unique methods to forecast myocardial infarction rates in England. Because of the complexity of the model, we have dedicated this paper just to reporting the forecast results from the microPRIME model. In future analyses we will conduct scenario analyses with implications for public health policies, but here the policy implications are restricted to the benefits of better forecasts of future disease burden. We already discuss this in the original manuscript (see example text below).

FROM THE ORIGINAL MANUSCRIPT: “Although we only modelled MI rates, due to similar risk factors for other major cardiovascular diseases (angina, heart failure and stroke) our results suggest that large increases in cardiovascular disease incidence in England are not likely before 2035. They also suggest that increases in prevalence may have already occurred and will now be followed by declines in men and steady levels for women, which has implications for NHS resource planning.”

We agree that the manuscript would benefit from a brief discussion of how we can improve modelling forecasts. We have added the following to the discussion:

“There are two developments that would increase the usefulness of non-communicable disease forecast models, such as microPRIME. First, forecast models would benefit from multiple disease outcomes across different categories of non-communicable diseases. Being able to forecast across multiple diseases has obvious benefits for resource planning, but it could also improve the accuracy of model forecasts for any given disease. This is because non-communicable diseases share common risk factors, so trends in these risk factors will affect disease incidence across multiple diseases – without explicitly incorporating these multiple disease pathways in the model framework a model will not account for related trends in these competing risks. Second, public health policymakers would benefit from models that incorporate more behavioural risk factors for disease (e.g. poor diet, lack of physical activity, alcohol consumption). By including such risk factors (either through direct associations with disease outcomes, or indirectly via body weight, blood pressure etc.) forecast models would be able to run policy scenarios aimed at changing behaviour.”

---

## [Editor Report · Decision Letter 1]

7 Jun 2022

Forecast of myocardial infarction incidence, events and prevalence in England to 2035 using a microsimulation model with endogenous disease outcomes

PONE-D-21-32355R1

Dear Dr. Scarborough,

We’re pleased to inform you that your manuscript has been judged scientifically suitable for publication and will be formally accepted for publication once it meets all outstanding technical requirements.

Kind regards,

Y Zhan

Academic Editor

PLOS ONE
---

## [Editor Report · Acceptance letter]

10 Jun 2022

PONE-D-21-32355R1 

Forecast of myocardial infarction incidence, events and prevalence in England to 2035 using a microsimulation model with endogenous disease outcomes 

Dear Dr. Scarborough:

I'm pleased to inform you that your manuscript has been deemed suitable for publication in PLOS ONE. Congratulations! Your manuscript is now with our production department. 

Kind regards, 

on behalf of

Dr. Y Zhan 

Academic Editor

PLOS ONE